# Comprehensive phase diagram of two-dimensional space charge doped $Bi_2Sr_2CaCu_2O_{8+x}$

Edoardo Sterpetti[1], Johan Biscaras[1], Andreas Erb[2] & Abhay Shukla [1]

The phase diagram of hole-doped high critical temperature superconductors as a function of doping and temperature has been intensively studied with chemical variation of doping. Chemical doping can provoke structural changes and disorder, masking intrinsic effects. Alternatively, a field-effect transistor geometry with an electrostatically doped, ultra-thin sample can be used. However, to probe the phase diagram, carrier density modulation beyond $10^{14}$ $cm^{-2}$ and transport measurements performed over a large temperature range are needed. Here we use the space charge doping method to measure transport characteristics from 330 K to low temperature. We extract parameters and characteristic temperatures over a large doping range and establish a comprehensive phase diagram for one-unit-cell-thick BSCCO-2212 as a function of doping, temperature and disorder.

[1] Institut de Minéralogie, de Physique des Matériaux et de Cosmochimie, Sorbonne Universités, Université Pierre et Marie Curie Paris 6, UMR CNRS 7590, MNHN, IRD UMR 206, 4 Place Jussieu, F-75005 Paris, France. [2] Walther Meissner Institut fur Tieftemperaturforschung, Bayerische Akademie der Wissenschaften, Walther-Meissnerstr. 8, 85748 Garching, Germany. Correspondence and requests for materials should be addressed to A.S. (email: abhay.shukla@upmc.fr)

High-temperature cuprate superconductors display a rich phase diagram, explored using a variety of techniques including transport[1–4], by varying chemical doping. Their parent compounds are known to be strongly correlated with antiferromagnetic ordering at low temperatures. As doping is increased this ordering temperature falls, eventually giving place to fluctuations, as superconductivity sets in at lower temperatures. The superconducting temperature increases with doping until an optimal value and then falls, following a dome-like shape, until superconductivity is suppressed. Meanwhile, at higher temperatures the non-superconducting state remains intriguing. In the underdoped regime, a pseudogap in the electronic excitation spectrum, possibly related to the formation of superconducting pairs, has been shown to exist. The temperature below which this pseudogap state opens decreases with doping. Near the optimal doping level and above the superconducting temperature, the material is thought to exist in a "strange" metallic phase in the sense of a deviation from Fermi liquid behavior as seen from the linear dependence of resistivity on temperature. Finally, as doping is further increased and superconductivity is suppressed, Fermi liquid behavior is progressively established.

As chemical doping is a source of disorder and structural change, alternative doping methods have been sought. To match the chemical doping levels with electrostatic doping, intense electric fields ~$10^9$ V·m$^{-1}$ and high carrier density modulation are required. In recent years, pioneering efforts have been made using ionic liquids[5, 6] or ferroelectrics as the dielectric to investigate the $La_{2-x}Sr_xCuO_4$ and $YBa_2Cu_3O_{7-x}$ families of materials[7–10]. Considerable change in the doping and critical temperature have been obtained and the insulator–superconductor transition in these materials has been investigated through transport measurements. However, for transport measurements the ionic liquid method is only relevant well below room temperature because of the conductivity of mobile ions in the liquid state. This precludes its use for establishing a temperature vs. doping phase diagram.

In the following, we use an alternative electrostatic doping method, namely, space charge doping[11, 12], achieving carrier density modulation up to $10^{15}$ cm$^{-2}$ and measure transport up to 330 K. We detect superconductivity in the two-dimensional limit in one-unit-cell-thick samples and investigate the effects of doping, temperature, and disorder to establish a comprehensive phase diagram of BSCCO-2212.

## Results

**Sample characterization.** BSCCO-2212 two-dimensional (2D) crystals with lateral size of ~100 μm and thickness ranging from 1 to 2.5 unit cells (u.c.) were fabricated by the anodic bonding method on 0.5 mm-thick soda-lime glass substrates[13, 14], using a bulk single crystal precursor with stoichiometry

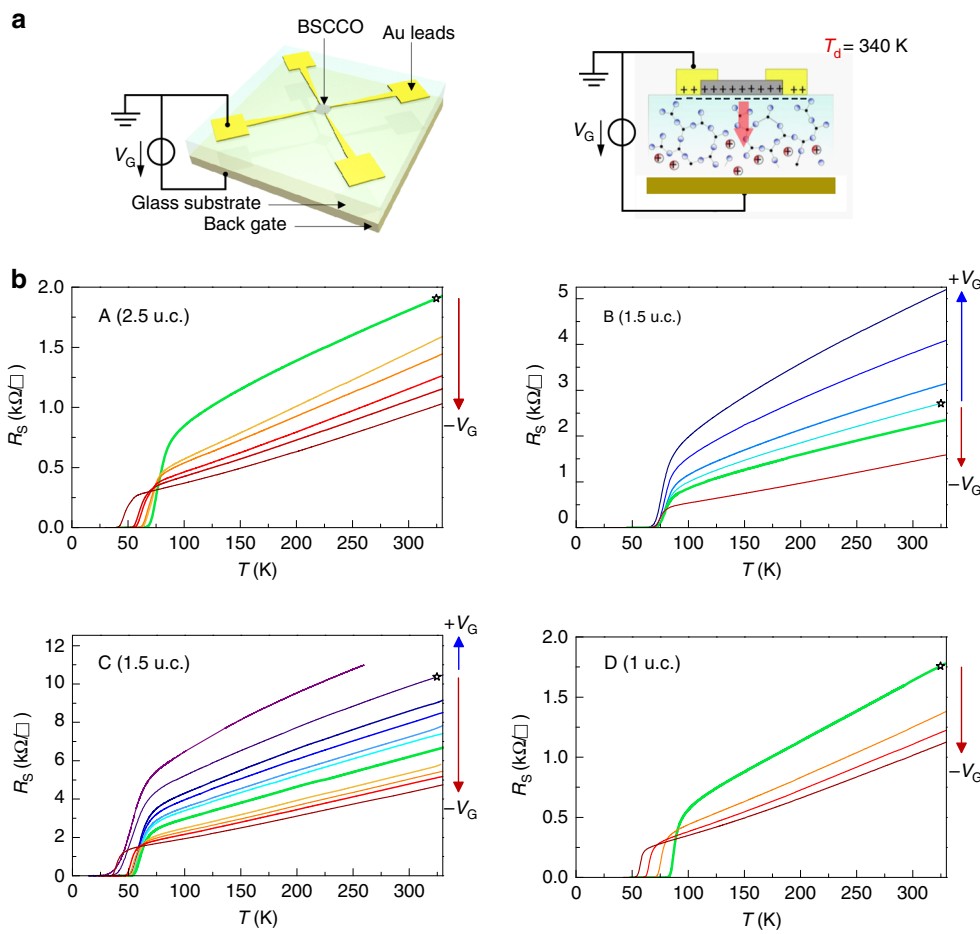

**Fig. 1** Overview of space charge doped BSCCO-2212 2D crystals. **a** Schematic top and side view of devices. **b** $R_S(T)$ for the four devices of this study at varying doping levels controlled by the gate voltage $V_G$. Curves corresponding to the initial doping are indicated with stars; the underdoped region corresponds to cyan-blue curves, whereas the overdoped region corresponds to yellow-brown curves; $R_S$ curves corresponding to the optimal doping level are plotted in thick green lines. Though samples A and D have not been swept through the optimal doping point, the fact that their initial doping is optimal is confirmed by the Hall coefficient measured in these samples as shown in Table 1

**Table 1 BSCCO-2212 device characteristics**

| Sample name | Thickness (nm) | $\mu_H$ (cm$^2$ V$^{-1}$ s$^{-1}$) | Initial doping (cm$^{-2}$) | $T_{c_{init}}$ (K) | $T_{c_{init}}$ (K) | $T_c(p_{opt})$ (K) | $T_c$ variation (K) |
|---|---|---|---|---|---|---|---|
| A | 7.0 (2.5 u.c.) | 9.4 | $8.0 \times 10^{14}$ | 65 | 38 | 65 | 27 |
| B | 4.5 (1.5 u.c.) | 8.7 | $7.0 \times 10^{14}$ | 70 | 59 | 71 | 12 |
| C | 4.5 (1.5 u.c.) | 2.8 | $3.8 \times 10^{14}$ | 30 | 18 | 51 | 33 |
| D | 3.2 (1.0 u.c.) | 10.7 | $8.3 \times 10^{14}$ | 81 | 51 | 81 | 30 |

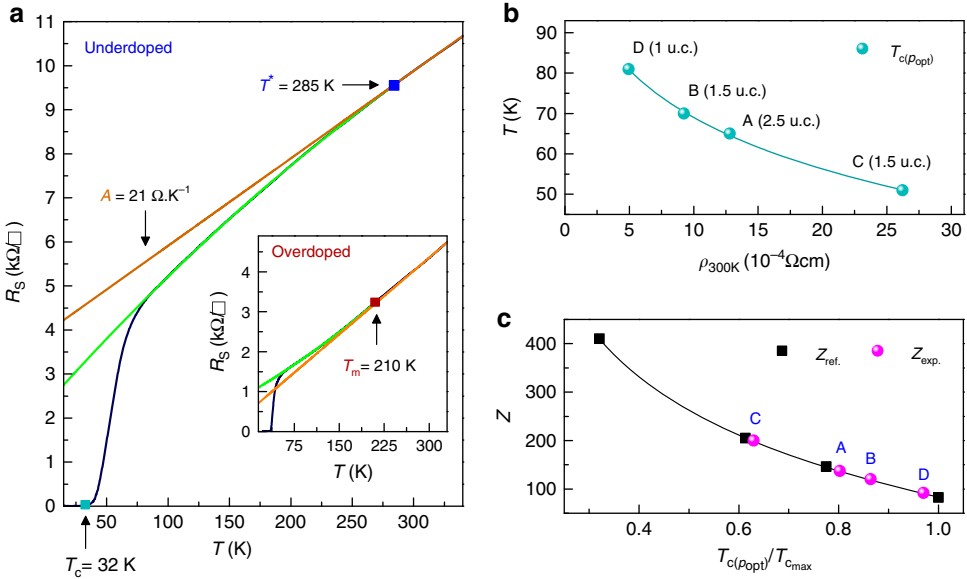

**Fig. 2** Extraction of characteristic temperatures and parameters. **a** Sheet resistance as function of temperature of an underdoped device (overdoped in the inset). **b** Maximum $T_c$ of each device as a function of resistivity; $T_c(p_{opt})$ is the $T_c$ at the optimal doping level or the peak of the superconducting dome. **c** Dependence of the scaling factor $Z$ on $T_c(p_{opt})/T_{c_{max}}$, where $T_{c_{max}} = 84$ K is the maximum $T_c$ measured by us to date in 2D BSCCO-2212 devices. The black line is a fit to the data (black squares) of the measurements of Naqib et al. ($Z_{ref.}$)[3], while the magenta dots are the calculated values for the four devices of this work ($Z_{exp.}$)

$Bi_{2.1}Sr_{1.9}CaCu_2O_{8+x}$ and a critical temperature of 89 K determined by magnetic susceptibility measurements. Contacts were deposited by evaporating 100 nm of gold through stencil shadow masks previously aligned on the sample. We thus avoid chemical pollution and surface contamination from standard lithography techniques, which in our experience can irreversibly degrade the quality of BSCCO-2212 2D crystals. We performed transport measurements with the four-wire Van der Pauw geometry in a high vacuum cryostat coupled to an external 2 T electromagnet for Hall measurements.

Here we show results for four different devices with thickness varying from 3 nm (1 u.c.) to ~7 nm (2.5 u.c.) as established by atomic force microscopy (AFM) and optical contrast. BSCCO-2212 cleaves at the BiO layer so that ultra-thin samples are expected to be multiples of 0.5 u.c. with one pair of $CuO_2$ planes per half unit cell. The thinnest samples fabricated were 0.5 u.c. and were systematically insulating. All devices shown in this work were superconducting at the outset, including the 1 u.c. device. The initial magnitude of resistivity varied, ranging from $5 \times 10^{-4}$ to $5 \times 10^{-3}$ Ω·cm depending on the device. We attribute this variation to the fabrication process, which can induce mechanical disorder caused by substrate roughness and eventually varying doping due to partial oxygen loss. At the start of the measurements, each device was annealed at 340–350 K (the temperature $T_d$ at which devices are doped) for a few hours at $10^{-6}$ mbar causing a decrease of the monitored sheet resistance $R_S$ probably due to removal of eventual surface contamination. An increase of $R_S$ would have indicated oxygen loss, not observed at

this temperature. We thus control carrier density exclusively by our electrostatic doping mechanism with no change in stoichiometry or disorder during the doping process.

**Space charge doping of 2D BSCCO-2212 crystals**. Doping is tuned by monitoring $R_S$ at the temperature $T_d$ by applying a gate voltage $V_G$ of the order of 100–280 V between the device and a metallic electrode on the opposite face of the glass substrate. The gate voltage creates an ionic space charge at the glass–device interface (Fig. 1a). Carrier concentration in the device can be fixed by lowering the temperature below $T_d$, thus freezing the accumulated space charge at the interface and annulling ionic mobility in the glass substrate (Supplementary Note 1). For a given carrier concentration, the temperature dependence of $R_S$ is measured from 330 K down to well below $T_c$, defined here as the temperature below which the resistivity is zero.

A central question is the evaluation of the carrier concentration ($n_S$). We perform Hall effect measurements and use $1/qR_H$ to approximately evaluate $n_S$ (Supplementary Note 2). This value is indicative of the change introduced by doping and not an absolute measure of the carrier density since the electronic structure at the Fermi surface and its evolution with doping and temperature is complex and cannot be approximated by a single parabolic band. The Fermi surface evolves from hole-like to electron-like with doping and the contributions of these surfaces to the Hall coefficient are temperature dependent[15–17]. To minimize errors related to this dependence, we consider $R_H$

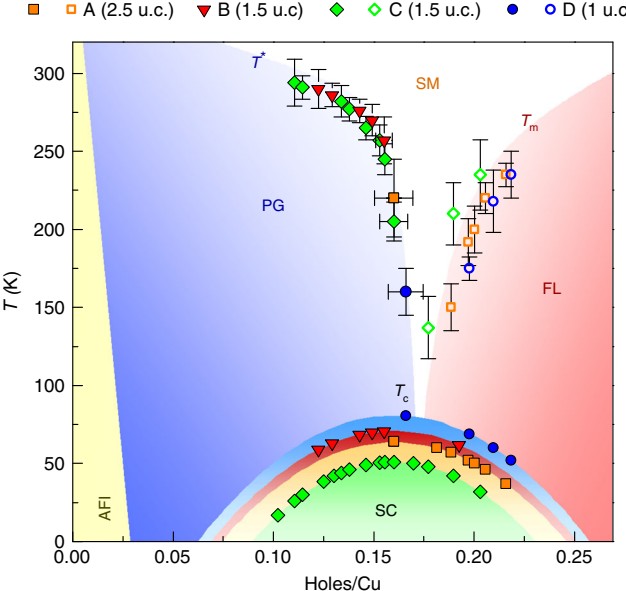

**Fig. 3** Phase diagram of 2D space charge doped BSCCO-2212. Of the 32 measured R$_S$(T) curves, 26 were of the quality needed for extracting characteristic temperatures. The characteristic temperatures $T^*$ (filled symbols), $T_m$ (open symbols), and $T_c$ (filled symbols on the domes) extracted for the four devices of this study are plotted as a function of doping (holes/Cu). They demarcate the boundaries of the pseudogap region (PG), the strange metal region (SM), the Fermi liquid region (FM), and the superconducting dome (SC). The different colors of the domes reflect the shrinking of the dome with disorder. The error on characteristic temperatures is estimated during their extraction while that on doping depends on $T_c$ through Eq. (1) and is significant around optimal doping. Where not shown, it is of the size of the data point or smaller

values at the fixed, low temperature of 120 K[18, 19] for all doping values. Figure 1 shows an overview of the R$_S$ vs. T characteristics as a function of space charge doping for the four devices of this study. We achieve a large, reversible variation in carrier density and hence in R$_S$ and $T_c$ by applying positive and negative V$_G$ to our devices, accessing both underdoped and overdoped regions of the phase diagram (Supplementary Note 3). At the outset, the intrinsic carrier concentration ($1/qR_H$) as well as the sheet resistance R$_S$ and the Hall mobility $\mu_H = R_H/R_S$ were measured for each sample. Table 1 shows these values along with the thickness, $T_{c_{init}}$ (initial $T_c$), $T_{c_{min}}$ (minimum $T_c$), $T_c(p_{opt})$ ($T_c$ at the optimal doping level) and variation in $T_c$ achieved through doping in each sample. In previous work, thin samples have been known to show higher R$_S$ and lower $T_c$ than thicker ones and superconductivity in 1 u.c. or thinner samples is only seen with graphene encapsulation[20–22]. Our unprotected 1 u.c. BSCCO-2212 devices are superconducting and we do not find a direct correlation between thickness and mobility leading us to conclude that the degree of disorder is the important parameter. This result can be attributed to our sample fabrication method[13, 14] as well as the avoidance of lithography during device fabrication (Supplementary Note 4). We note that disorder can have significant impact on superconducting properties including $T_c$ suppression[23–25], localization of Cooper pairs[26] and broadening of the superconducting transition due to spatial inhomogeneity of the superconducting condensate[27, 28].

**Phase diagram.** The R$_S$(T) curves obtained at various doping levels depend on the electronic structure of the corresponding phase. Variations in the temperature dependence of R$_S$ have been

successfully used to demarcate different regions of the cuprate phase diagram[1, 2, 29]. It is known that the normal state in the cuprates around optimal doping is characterized by a linear dependence of resistivity on temperature, generally called the "strange metal" phase. At higher doping and lower temperatures, this dependence assumes the normal metallic power law dependence in the "Fermi liquid" phase. At lower doping and lower temperatures, the dependence again deviates from linearity in a different manner in the region known as the "pseudo-gap" phase. The temperatures corresponding to deviations in linearity (Fig. 2a) can be used to determine the domain of existence of the different phases. It is to be seen if a single, coherent, doping vs. temperature phase diagram of BSCCO-2212 emerges from these measurements of devices with differing intrinsic disorder and $T_c$.

For each R$_S$(T) curve, we use a linear fit for the "strange metal" high temperature region (290 K < T < 330 K) with $\rho(T) = \rho_0 + AT$, where $\rho_0$ is the residual square resistivity and $A$ is the slope[1]. $T_m$ (temperature of crossover into the Fermi liquid phase) and $T^*$ (temperature of crossover into the pseudo-gap phase) are determined, respectively, as the temperature of upward and downward deviation (green curves) from the linear behavior (orange straight lines). Where possible, similar estimations were found for $T_m$ and $T^*$ by identifying the temperature at which d$R_s$/dT deviates downward (upward) from its high temperature constant value (Supplementary Note 5). The downward deviation below $T^*$ in BSCCO-2212 is ascribed to the reduction of inelastic scattering rate of electrons in the pseudo-gap phase, when temperature is lowered[30]. This characteristic $T^*$ may however be different[2] from the temperature of crossover into the pseudo-gap phase $T_{pg}$, commonly detected by photo-emission, Raman, or tunneling spectroscopy[17, 31, 32]. It is well established in cuprates that in the Fermi liquid phase the upward deviation of R$_s$(T) below $T_m$ follows the power law $\rho(T) = \rho_0 + BT^m$, where the exponent $m$ is expected to increase with doping from 1 to 1.5–2 depending on the material[3, 4, 33]. In our overdoped devices, we observed an $m(p)$ dependence[3], with a maximum $m = 1.35$ at the doping level of $1.6 \times 10^{15}$ cm$^{-2}$ (~0.21 holes/Cu). The accessible range (up to 330 K) where linear behavior is found at high temperature ensures a reliable determination of $T^*$ and $T_m$. The construction of the phase diagram for BSCCO-2212 also requires the determination of the carrier concentration for each device. We use two methods, the measured Hall coefficient ($1/qR_H$) and an estimation of doped holes (p) per Cu atom (holes/Cu) obtained by an empirical and commonly used $T_c(p)$ relation[34].

$$T_c(p)/T_c(p_{opt}) = 1 - Z(p - p_{opt})^2 \tag{1}$$

where $T_c(p_{opt})$ is the maximum critical temperature measured for each device corresponding to the optimal doping level and $Z$ is a scale factor. In bulk material with minimal defects, $Z$ is empirically determined to be 82.6[4, 35, 36] and superconductivity exists in the region $p = 0.05$ to $p = 0.27$ holes/Cu. Disorder reduces $T_c$ and constrains superconductivity to a smaller region in $p$[3] with a scaling of the superconducting dome. The lowering of $T_c$ due to disorder also corresponds to a higher normal state R$_S$(T), ascribed both to Coulomb interactions and scattering by impurities[23]. In our 2D crystals, this behavior can be expected as shown qualitatively in Fig. 2b. The empirical relation for obtaining $p$ from $T_c$ is modified in the presence of disorder[3] by modifying the scaling parameter $Z$ as shown in Fig. 2c. Thus characteristic temperatures are obtained as a function of doped holes/Cu allowing us to construct phase diagrams for our samples with varying intrinsic disorder.

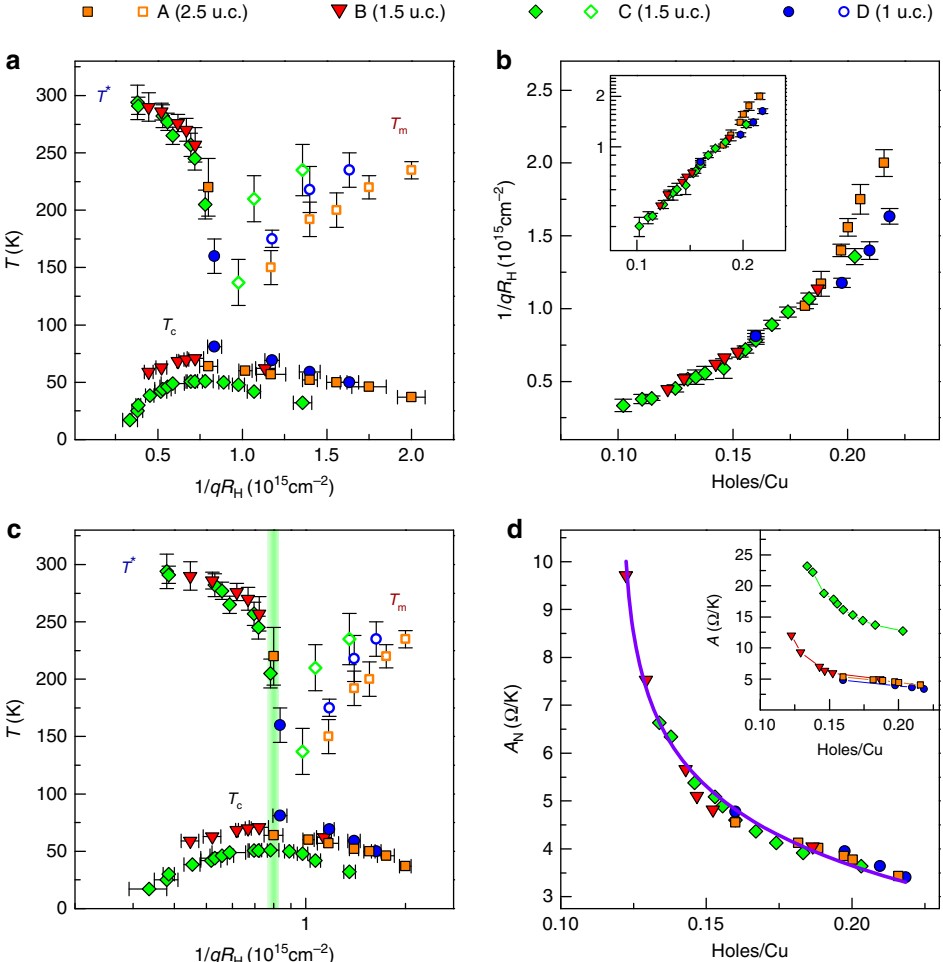

**Fig. 4** Phase diagram with carrier density inferred from $R_H$. **a** Characteristic temperatures as in Fig. 3 with $T^*$ (filled symbols), $T_m$ (open symbols) and (filled symbols below 100 K) of the four devices as a function of $1/qR_H$, and the resulting phase diagram. **b** Doping calibration: Exponential dependence of $1/qR_H$ on the calculated holes/Cu. **c** Same as **a** but accounting for the exponential functional dependence found in **b**. The vertical green band indicates the region of optimal doping. Though limitations in the use of $R_H$ for estimating doping preclude quantitative conclusions, a qualitative similarity with Fig. 3 can be seen. **d** Doping dependence of normalized slope $A_N$ of the linear part of $R_S(T)$ in the strange metal phase ($T > T^*, T_m$). $A_N$ values for the four devices fall on a single curve (solid line) given by $A_N \propto -b \times \ln(\text{holes/Cu})$. Inset: doping dependence of the slope $A$ before normalization. Error bars in **a**–**c** take into account uncertainty of determination of characteristic temperatures and $R_H$

## Discussion

The resulting phase diagram with doping $p$ determined from $T_c$ by Eq. (1) is shown in Fig. 3. The characteristic temperatures ($T_m$ and $T^*$) and their dependence on $p$ for the four devices establish remarkably coherent and well-demarcated domains corresponding to the well-known cuprate phase diagram. The superconducting dome, as determined from the empirical formula of Eq. (1), shrinks with increasing disorder as $T_c$ is suppressed, recalling the effect of a magnetic field[37-39]. We thus establish a complete phase diagram for ultra-thin BSCCO-2212 extending well into the underdoped and overdoped regions and explored entirely using electrostatic doping. In particular, we provide data in the overdoped region where scant information exists for BSCCO-2212[35, 40]. The extracted phase boundary, notably between pseudogap region and the strange metal region, is not linear. Extrapolating this phase boundary to the doping axis would indicate a critical doping close to the optimal value, but higher values can be obtained by extrapolating segments of the phase boundary on the underdoped side. Finally, we construct the same phase diagram using carrier density estimated by $1/qR_H$ (Fig. 4a) and reach two conclusions. First, the relation between $p$ determined using Eq. (1) and $1/qR_H$ is non-linear. This is

confirmed in Fig. 4b where an exponential dependence is found between the two quantities[16, 19] and in Fig. 4c where the horizontal $1/qR_H$ axis is plotted in logarithmic scale yielding a result very similar to the phase diagram obtained in Fig. 3. However, we must keep in mind the limitations in the use of $R_H$ for estimating doping discussed above and note only the qualitative similarity between the phase diagrams of Figs 3 and 4c. Second, the data corresponding to the 2.5 u.c. device appears incoherent with the other devices while this was not the case in Fig. 3, where $p$ is determined from the $T_c$ using Eq. (1). The probable reason is that in our 2.5 u.c. device the doping is not homogeneous over the total thickness given the small Thomas–Fermi screening length of cuprates ($\lambda_{TF} \sim 1$ u.c.)[8, 41] introducing an error in the determination of carrier density from $1/qR_H$. We conclude our analysis of $R_S(T)$ as a function of doping by examining the normal state behavior in greater detail. In the normal state, it has been shown that the slope $A$ of the linear part of the $R_S(T)$ curve is inversely proportional to the doping with a simple, universal $1/p$ power law behavior[1, 42]. We confirm this finding as shown in Fig. 4d where we have accounted for the varying disorder in our devices by normalizing the measured slope $A$ of the $R_S(T)$ curve by $A_N = A\mu_H/\mu_{H_{max}}$, where $\mu_H$ is the measured mobility of each device at

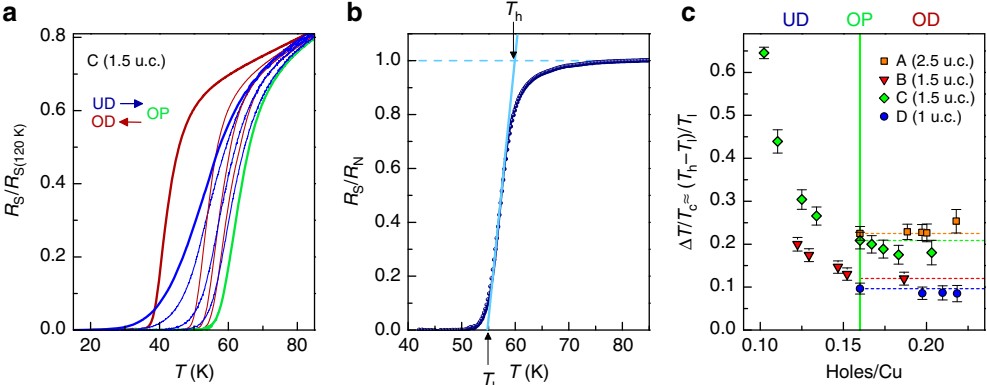

**Fig. 5** Transition width dependence on electrostatic doping. **a** $R_S(T)$ of device C (1.5 u.c.), normalized to $R_S$(120 K). Thick lines indicate the most underdoped (UD, blue), the optimally doped (OP, green), and the most overdoped (OD, brown) curves. **b** $R_S(T)/R_N(T)$ (with $R_N(T)$ normal state resistance) in the proximity of the superconducting transition. $R_N(T)$ is determined as $R_0 + BT^m$ or as $R_0 + aT - bT^2$ according to whether the device is overdoped or underdoped, respectively. A linear component is fitted to the superconducting transition with $T_h$ and $T_l$ extrapolated as the intersection with the dashed green line and the x axis, respectively. **c** Doping dependence of the normalized superconducting transition width $\Delta T/T_c$, for varying thickness and disorder. Error bars take into account uncertainty of determination of transition width

$T = 120$ K and $\mu_{H_{max}}$ is the highest measured mobility of our devices (in sample D, 1 u.c.).

While the superconducting transition is by definition abrupt, in 2D systems a smooth transition occurs whose nature is thought to belong to the Kosterlitz–Thouless (KT) class[43–46] and which is characterized by the temperature $T_{KT}$ above which superfluid density discontinuously attains zero[47, 48]. Such a transition is difficult to observe experimentally because of competing phenomena such as disorder[27] and finite-size effects[49], which broaden the superconducting transition for $T < T_{KT}$. These factors induce spatial inhomogeneities of the superconducting condensate on the mesoscopic scale[50] affecting critical temperatures and the superconducting gap[51]. Though an in-depth analysis of the influence of dimensionality, disorder and doping is beyond the scope of this work, we probe below the variation of the superconducting transition width as a function of doping.

In Fig. 5a, the superconducting transition from the underdoped to the overdoped region is shown for device C (1.5 u.c.). A strong reduction of the transition width with doping is clearly visible. Since doping affects both the superconducting transition width and the normal state behavior, we first normalize each $R_S(T)$ curve by $R_N(T)$, its normal state component, fitted by $R_N(0) + BT^m$ in the overdoped region, or by $R_N(0) + aT - bT^2$ in the underdoped region. To estimate an intrinsic superconducting transition width, we must eliminate extrinsic contributions (e.g., from disorder), which tend to broaden the transition either at the onset or near $T_c$. This is done using the normalized $R_S(T)/R_N(T)$ curves by fitting a linear component to the transition and extracting two limiting temperatures $T_h$ and $T_l$, corresponding to the "intrinsic" onset and critical temperature as shown in Fig. 5b. The transition width normalized to the critical temperature is then approximated as $\Delta T/T_c \approx (T_h - T_l)/T_l$. In conventional 2D superconductors, $\Delta T/T_c \simeq (\alpha_0/\xi)^2$ (where $\alpha_0$ is the inter-atomic distance and $\xi$ the coherence length) is of the order of 0.02 and the broadening of the superconducting transition is ascribed to Aslamazov–Larkin fluctuations[28]. Fluctuation phenomena are prominent in cuprates due to the short coherence length and $\Delta T/T_c$ is expected to be higher. In Fig. 5b, we show the variation of $\Delta T/T_c$ for each device as a function of doping $p$. It is significantly higher than for a conventional 2D superconductor and decreases strongly with doping in the underdoped region. We remark that devices A (2.5 u.c.) and C (1.5 u.c.) have a higher $\Delta T/T_c$. Device C (1.5 u.c.) has the lowest mobility and is probably the most

disordered while the thickest device A (2.5 u.c.) can be expected to have a gradient in the doping over the total thickness (Supplementary Note 6). Thus extrinsic factors like disorder and inhomogeneity can lead to an increased $\Delta T/T_c$. However, in all devices $\Delta T/T_c$ decreases strongly with doping until the optimal doping is attained and then remains almost unchanged. This can be ascribed to a change in the inhomogeneity landscape[49, 52], which in turn increases the characteristic dimension of homogeneous superconducting domains and the spatial coherence of the superconducting state. Electronic correlations are also reduced as carrier density and coherence increase.

In conclusion, we fabricate ultra-thin, superconducting (1–2.5 unit cell) BSCCO-2212 devices to which we apply a highly efficient electrostatic doping technique called space charge doping. Starting from the nominal optimal doping, we obtain carrier density variations of the order of $10^{15}$ cm$^{-2}$, as estimated by Hall measurements, both in the overdoped and in the underdoped regions, entirely through space charge doping. We measure sheet resistance as a function of doping from 330 K to well below the superconducting transition. From these measurements, we extract characteristic temperatures demarcating the various regions of the phase diagram. These measurements allow us to construct a comprehensive phase diagram as a function of doping, temperature, and disorder in 2D $Bi_2Sr_2CaCu_2O_{8+x}$ and to analyze the influence of doping, disorder, and two dimensionality. The variation of normal state characteristics of the sheet resistance as well as the variation of the superconducting transition width is traced continually as a function of doping. Our results demonstrate the potential of space charge doping in 2D crystals to develop the fundamental understanding of phase transitions in these materials as well as possible applications.

## Methods

**Sample preparation.** Few-nm BSCCO-2212 samples were made by the anodic bonding method[13, 14]. $Bi_2Sr_2CaCu_2O_{8+x}$ flakes were peeled off from bulk crystals and deposited on a 0.5 mm-thick soda-lime glass substrate with 1 nm roughness according to AFM topography measurements. Substrate and precursor were subsequently placed between two electrodes and heated between 150 and 200 °C in order to activate Na$^+$ mobility within the glass. High negative voltage was then applied to the face of the glass opposite of the sample to attract Na$^+$ ions, thus creating an O$^{2-}$ space charge at the glass–sample interface. After 5–10 min, the precursor is electrostatically bonded to the glass and the voltage can be removed. This electrostatic bond involves the first few nm of the precursor, and using adhesive tape, the upper layers can be mechanically exfoliated, leaving large area

ultra-thin BSCCO-2212 on the glass substrate. The thickness of the samples were evaluated by AFM measurements and optical contrast.

**Measurements details.** Electrical contacts were made in Van der Pauw geometries by depositing 100 nm Au in a standard thermal evaporator at the pressure of $10^{-6}$ mbar. The backside of the glass substrate was then glued by silver epoxy to a gold electrode evaporated on top of an insulating MgO substrate to act as a back gate. Four point resistivity and Hall measurements were made in a high vacuum ($10^{-6}$ mbar) Oxford He-flow cryostat with a minimum temperature of 2.8 K and maximum 420 K. A resistive electromagnet was used to apply magnetic fields up to 2 T perpendicular to the sample.

**Data availability.** The data that support the findings of this study are available from the corresponding authors upon request.

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

## Acknowledgements

We thank the Institut des NanoSciences de Paris for access to the electromagnet facility. We acknowledge the Consortium des salles blanches d'Ile de France, M. Rosticher and J. Palomo for access to clean room facilities. This work was supported by French state funds managed by the ANR within the Investissements d'Avenir programme under reference ANR-11-IDEX-0004-02 and more specifically within the framework of the Cluster of Excellence MATISSE led by Sorbonne Universités.

## Author contributions

A.S. designed the project. A.E. synthesized the BSCCO single crystal precursors, E.S. fabricated the devices, performed the measurements with J.B. and analyzed the data. E.S., J.B., and A.S. wrote the paper.

## Additional information

**Competing interests:** The authors declare no competing financial interests.

