## [Peer Review File · Nature Communications]

Reviewers' comments:

Reviewer #1 (Remarks to the Author):

Sterpetti et al review

The authors have exploited the space-charge doping technique to vary the carrier concentrations of unit-cell thick and near unit-cell thick flakes of $\text{Bi}_2\text{Sr}_2\text{CaCu}_2\text{O}_{8+x}$, and to develop a phase diagram of the compound, delineating the superconducting, pseudogap, strange metal and Fermi liquid regimes. The result qualitatively resembles the phase diagram obtained by the laborious process of chemical doping. The measurements were carried out on flakes that were anodic-bonded to soda-lime glass wafers. Space-charge was induced at the flake-glass interface by applying voltage between the flake on one face of the glass wafer and a gate on the opposite face while subjecting the combination to an elevated temperature of 340K, sufficient to allow the sodium ions of the glass wafer to migrate. The number of holes could either be increased or decreased relative to the hole concentration of the pristine flake depending upon the sign of the bias voltage. Resistance vs. temperature curves at different hole concentrations were used to determine the hole concentration dependencies of the superconducting transition temperature, and the transition temperatures of the crossovers from the strange metal phase to the Fermi liquid phase, and from the strange metal to the pseudogap phase. The dependence of the width of the superconducting transition on hole concentration was also reported. Measurements were carried out on four samples of various thicknesses (1u.c, 1.5 u.c., 1.5 u.c., and 2.5 u.c.), the results of which were aggregated to construct the phase diagram. The hole concentrations, p per Cu atom (holes/Cu), were determined in two ways, using the measured Hall coefficient, and employing an empirical and commonly used empirical relationship between the superconducting transition temperature and p . The former was found to vary exponentially with the latter. When this was taken into account the phase diagram with temperatures plotted vs p determined from the empirical formula was found to be identical with that plotted vs. the logarithm of p determined from the Hall coefficient, as it should have been.

The work described in this manuscript is of high quality and involves a unique and extremely useful electrostatic gating technique. It suffers from a problem common to many such techniques when applied to cuprate superconductors, the difficulty in determining the carrier concentration, because of the nontrivial variation of the Hall effect with temperature and carrier concentration and the uncertainty as to the general applicability of the empirical relationship between the transition temperature and the carrier concentration. Nevertheless, I recommend publication in Nature Communications.

There are a couple of issues that should be addressed before publication. First, the phase diagram is convincing when the results from the four distinct samples are aggregated, but not too convincing when each is considered separately. The problem is that there may be an uncertainty regarding two of the samples, A and D, which are subjected to only negative gate biases, which add holes. Samples B and C are subjected to both signs of bias, so that holes are added and subtracted. For these samples, this permits one to experimentally identify the optimal doping level as that resulting in the highest superconducting transition temperature. This identification is crucial to determining the hole doping from the empirical formula. It turns out that for both samples B and C, the transition temperature of the ungated flake with the initial doping, is not the highest and does not correspond to the optimum doping level. However, for samples A and D, which are only gated with one sign of the bias, the initial doping is taken to be the optimal doping. Without the other sign of gate bias, one cannot be certain that this is true. If it were not true, then the dependencies of the various crossover and transition temperatures on hole concentration for these two samples would be different from those which are currently shown, and the basis for claiming a determination of the phase diagram would be quite thin. The authors should address this issue.

Figures 3 and 4(a) which show the various temperatures vs. hole concentration determined using

the two methods are quite different, which would be expected given the data of Fig. 4(b). However, it is obvious that Fig 4(c) should resemble Fig. 3 because of the relationship found in Fig. 4(b). This raises an issue as to whether there might be another way to determine p - that might be more reliable. Would it be possible to integrate the gate current during the charging process? This should give at least an upper bound on the charges induced in the samples. Presumably one would have to insulate the electrical leads to the sample flake in order for this to work. Have the authors considered this?

Finally, there is the measurement of the superconducting transition width-dependence on electrostatic doping. The results are interesting, but in the absence of a detailed analysis based on the theory of Refs. 25 and 49, they are really not terribly useful. The authors might consider dropping this discussion or shortening it in the final version of the manuscript. This type of measurement is really not relevant to the main focus of the manuscript, which is on the phase diagram as referred to in the title and abstract.

Reviewer #2 (Remarks to the Author):

This paper reports on a comprehensive doping-temperature dependence study of the transport properties of Bi2212 ultrathin flakes using the so-called space charge doping technique the authors previously invented. This technique (which I would call ionic solid gating instead) can achieve carrier density modulation of the same order as the ionic liquid gating, while not having the major problem due to unwanted chemical reaction. It definitely has an enormous potential for applications as the authors have previously demonstrated on graphene and MoS₂ and, now, on Bi2212. The results obtained on Bi2212 in terms of the normal-state resistance and superconducting transition width look impressive. In particular, the presented phase diagram appears to point to the presence of a quantum critical point close to the optimal doping level. It looks quantitatively different from that obtained from bulk single crystals [Proc. Natl. Acad. Sci. 109, 18332 (2012)], thereby provoking thoughts about the roles of non-electronic degrees of freedom (disorder, lattice) in shaping the generic phase diagram of cuprates. This paper is therefore a valuable contribution to high- T_c field and has a broad appeal beyond that field. I recommend its publication in Nature Communications provided improvements be made in its revision on the following aspects.

1. A major challenge could be raised regarding the stability of space charge density during the temperature dependence measurement. Fig. 1 in the supplement does not show any sign of saturation in sheet resistance upon lowering temperature toward and close to 300 K, implying that the freezing of space charge might not be as effective as claimed. This issue is critical for correctly assessing the temperature-dependent result obtained at a given bias voltage in terms of whether it corresponds to a fixed doping level.
2. The presented phase diagram appears to point to the presence of a quantum critical point close to the optimal doping level. This is quite different from the bulk single crystal results that highlight a critical doping (if existing) close to 19 % and a pseudogap phase boundary of a possibly non-trivial form. Given the potential significance of this result and the uncertainty in the doping level estimation, related discussions are warranted.
3. In order to set up a benchmark for future studies using the same doping technique, it is necessary to provide more details and further elucidations about the technique. Whether or not some of them can be found in earlier papers by the same authors, it helps to make the paper self-contained.

3a. Is it purely electrostatic in nature? Any sign of irreversibility or hysteresis observed in experiment that might suggest a partial electrochemical character of the doping method?
3b. What is the thickness of the glass substrate? Why is the applied bias voltage so large compared to the liquid gating case? Is that because the glass substrate at high temperatures has a much worse metallicity than the ionic liquid?
3c. What is the effective thickness of the sample that has been charged for "thick" films (say >1 μc)? Since the screening length is not a priori known, this may bring in a big uncertainty in the data analysis. For example, for an overdoped cuprate, the whole film is conducting even without gating. Then after gating, its resistivity changes. But if we don't know the thickness of the film that has been charged, there is no way to separate the contributions of the charged and uncharged portions of the film and get a solid answer of how resistivity changes during gating.

Ruihua He

Reviewer #3 (Remarks to the Author):

In this manuscript, the authors used space charge doping (or others may call it solid electrolyte gating) to tune thin Bi₂Te₂Se flakes with thickness from 1 to 2.5 unit cells. For samples with different doping level, they extracted characteristic temperatures from transport measurements and constructed a phase diagram. The research is highly interesting and valuable for people working on unconventional superconductivity and beyond. My comments are detailed below, and I would suggest a minor revision before the manuscript can be accepted for publication.

1. With sample B and C having the same thickness, the maximum attainable transition temperature varied by 20K, the authors suggested that it is due to effects of disorder, which is reasonable. Disorder seems also one of the reasons that they failed to observe superconductivity in samples with 0.5 unit cell thick. Another possible reason is the loss of carriers in such an ultra-thin sample after exfoliation, due to oxygen release or reconstruction in the Bi-O plane. It might be possible to apply different gate voltages and make measurements down to even lower temperature than 2.8K. Can the authors make such measurements or make comments on that?

2. With electrolyte gating, it was already reported that small ions like Li⁺ or Na⁺ may get intercalated into gaps in the lattice (in this case it is the space between the Bi-O layers). Possible evidences include irreversible gating behavior or sudden change of resistivity during gating. The authors shall give experimental evidences excluding this possibility, because otherwise, the interpretation of the data would become much more complicated.

3. As the authors have suggested, disorder plays an important role in such ultra-thin sample where the exposed surface may become very reactive, ready to have physical or chemical absorption or subject to surface reconstruction. While 1 unit cell Bi₂Te₂Se was reported to be insulating (ref. 18), and in this manuscript, it is superconducting. Is there any control of the sample storage or measurement environment? Did the authors observed any degradation in samples in less controlled environment?

4. I would suggest that the authors add more data to table 1 to make it more complete. For example, initial T_c, maximum and minimum T_c and the gate voltages on which they were obtained. Although many of these information are available in figure 1, however, it would be good if the authors summarize these results quantitatively in the table.

Reply to reviewers

Below we reply to all points raised by the reviewers. We wish to thank all reviewers for their appreciation of our work, the discussion and the pertinent points raised. Our replies are in blue and in the revised manuscript and supplementary material the revisions have been signalled in red.

Reviewer #1 (Remarks to the Author):

Sterpetti et al review

The authors have exploited the space-charge doping technique to vary the carrier concentrations of unit-cell thick and near unit-cell thick flakes of $\text{Bi}_2\text{Sr}_2\text{CaCu}_2\text{O}_{8+x}$, and to develop a phase diagram of the compound, delineating the superconducting, pseudogap, strange metal and Fermi liquid regimes. The result qualitatively resembles the phase diagram obtained by the laborious process of chemical doping. The measurements were carried out on flakes that were anodic-bonded to soda-lime glass wafers. Space-charge was induced at the flake-glass interface by applying voltage between the flake on one face of the glass wafer and a gate on the opposite face while subjecting the combination to an elevated temperature of 340K, sufficient to allow the sodium ions of the glass wafer to migrate. The number of holes could either be increased or decreased relative to the hole concentration of the pristine flake depending upon the sign of the bias voltage. Resistance vs. temperature curves at different hole concentrations were used to determine the hole concentration dependencies of the superconducting transition temperature, and the transition temperatures of the crossovers from the strange metal phase to the Fermi liquid phase, and from the strange metal to the pseudogap phase. The dependence of the width of the superconducting transition on hole concentration was also reported. Measurements were carried out on four samples of various thicknesses (1u.c, 1.5 u.c., 1.5 u.c., and 2.5 u.c.), the results of which were aggregated to construct the phase diagram. The hole concentrations, p per Cu atom (holes/Cu), were determined in two ways, using the measured Hall coefficient, and employing an empirical and commonly used empirical relationship between the superconducting transition temperature and p . The former was found to vary exponentially with the latter. When this was taken into account the phase diagram with temperatures plotted vs p determined from the empirical formula was found to be identical with that plotted vs. the logarithm of p determined from the Hall coefficient, as it should have been.

The work described in this manuscript is of high quality and involves a unique and extremely useful electrostatic gating technique. It suffers from a problem common to many such techniques when applied to cuprate superconductors, the difficulty in determining the carrier concentration, because of the nontrivial variation of the Hall effect with temperature and carrier concentration and the

uncertainty as to the general applicability of the empirical relationship between the transition temperature and the carrier concentration. Nevertheless, I recommend publication in Nature Communications.

There are a couple of issues that should be addressed before publication.

1) First, the phase diagram is convincing when the results from the four distinct samples are aggregated, but not too convincing when each is considered separately. The problem is that there may be an uncertainty regarding two of the samples, A and D, which are subjected to only negative gate biases, which add holes. Samples B and C are subjected to both signs of bias, so that holes are added and subtracted. For these samples, this permits one to experimentally identify the optimal doping level as that resulting in the highest superconducting transition temperature. This identification is crucial to determining the hole doping from the empirical formula. It turns out that for both samples B and C, the transition temperature of the ungated flake with the initial doping, is not the highest and does not correspond to the optimum doping level. However, for samples A and D, which are only gated with one sign of the bias, the initial doping is taken to be the optimal doping. Without the other sign of gate bias, one cannot be certain that this is true. If it were not true, then the dependencies of the various crossover and transition temperatures on hole concentration for these two samples would be different from those which are currently shown, and the basis for claiming a determination of the phase diagram would be quite thin. The authors should address this issue.

Reply: This is an important point. First the reason why samples A and D were doped only on the overdoped side: as we have stated in the manuscript, the data presented here on BSCCO is rare even with chemical doping and even more so on the overdoped side (hole doping for an optimally doped sample). We thus wanted good data sets for this side of the phase diagram. As samples remain delicate and tend to 'die' after a certain amount of measurements, we decided to map this side of phase diagram as completely as possible. The decision to preferentially measure one side of the phase diagram for these two samples was taken because measurements on samples B and C showed that the n_s estimated using the 120K Hall coefficient is $\sim 8 \cdot 10^{14} \text{ cm}^{-2}$ for optimally doped BSCCO. This can also be seen in Table 1 which shows that A, B and D are very similar in that the initial doping is optimal or near optimal, while C is clearly not. This fact is explained because A, B and D were measured immediately after sample and device fabrication (a duration of about 2-4 days in primary vacuum except when being processed), while sample C was stored in primary vacuum for one month before measurements, probably resulting in oxygen loss during storage. The measured n_s is a reliable indicator for optimal doping. This is graphically verified in Fig 4c where the phase diagram is plotted as a function of the inverse Hall coefficient. If the initial doping was different from near optimal, the data of samples A and D would not be coherent. We have added the following sentence in the manuscript in the caption of Figure 1 and thank the referee for raising this point.

Though samples A and D have not been swept through the optimal doping point, the fact that their initial doping is optimal is confirmed by the Hall coefficient measured in these samples as shown in Table 1.

2) Figures 3 and 4(a) which show the various temperatures vs. hole concentration determined using the two methods are quite different, which would be expected given the data of Fig. 4(b). However, it is obvious that Fig 4(c) should resemble Fig. 3 because of the relationship found in Fig. 4(b). This raises an issue as to whether there might be another way to determine p - that might be more reliable. Would it be possible to integrate the gate current during the charging process? This should

give at least an upper bound on the charges induced in the samples. Presumably one would have to insulate the electrical leads to the sample flake in order for this to work. Have the authors considered this?

Reply: This is a point on which we have spent some time and effort. Estimating charge transfer by integrating current would be useful but is not currently possible. We have made systematic calibration measurements with a standard sample and Hall measurements to try and accomplish this. Two difficulties have prevented us from arriving at a result. Firstly the typical sample ($10^4 \mu\text{m}^2$) represents a small fraction of the area where charge is transferred when compared to the contact leads and pads (few mm^2). This difficulty could in principle be overcome by assuming uniform charging and very precise gate current measurement. However minute current leaks which persist in the path through the cryostat between the electronics and the sample, despite insulated leads, make this difficult at present. A total rewiring may help, but we cannot be sure. In the meanwhile the Hall measurements are reliable, if not quantitatively exact in this case.

We have added the following sentence to Supplementary note 1 which discusses the space charge doping method:

Estimating charge transfer by integrating the gate current would be useful but is not currently possible because of experimental constraints.

3) Finally, there is the measurement of the superconducting transition width-dependence on electrostatic doping. The results are interesting, but in the absence of a detailed analysis based on the theory of Refs. 25 and 49, they are really not terribly useful. The authors might consider dropping this discussion or shortening it in the final version of the manuscript. This type of measurement is really not relevant to the main focus of the manuscript, which is on the phase diagram as referred to in the title and abstract.

Reply: We agree with the general point of the reviewer. We have more detailed measurements (and still more in progress) as well as relevant fits to these which will form the basis of another publication. However as the reviewer says, we found that a small preview is in effect interesting, even though it is not the main thrust of the paper. We would be much obliged if the reviewer allows us to retain this part.

Reviewer #2 (Remarks to the Author):

This paper reports on a comprehensive doping-temperature dependence study of the transport properties of Bi2212 ultrathin flakes using the so-called space charge doping technique the authors previously invented. This technique (which I would call ionic solid gating instead) can achieve carrier density modulation of the same order as the ionic liquid gating, while not having the major problem due to unwanted chemical reaction. It definitely has an enormous potential for applications as the authors have previously demonstrated on graphene and MoS2 and, now, on Bi2212. The results obtained on Bi2212 in terms of the normal-state resistance and superconducting transition width look impressive. In particular, the presented phase diagram appears to point to the presence of a quantum critical point close to the optimal doping level. It looks quantitatively different from that obtained from bulk single crystals [Proc. Natl. Acad. Sci. 109, 18332 (2012)], thereby provoking thoughts about the roles of non-electronic degrees of freedom (disorder, lattice) in shaping the generic phase diagram of cuprates. This paper is therefore a valuable contribution to high-Tc field and has a broad

appeal beyond that field. I recommend its publication in Nature Communications provided improvements be made in its revision on the following aspects.

1. A major challenge could be raised regarding the stability of space charge density during the temperature dependence measurement. Fig. 1 in the supplement does not show any sign of saturation in sheet resistance upon lowering temperature toward and close to 300 K, implying that the freezing of space charge might not be as effective as claimed. This issue is critical for correctly assessing the temperature-dependent result obtained at a given bias voltage in terms of whether it corresponds to a fixed doping level.

Reply: Below 320K, ion mobility as estimated through gate current is negligible, so this discussion is limited to the temperature range 320-340K. The sheet resistance varies with temperature as well as doping so in the supplementary Fig1 (which includes a longer time scale now) the relevant quantity to be monitored is the gate current I_G . It can be seen that when the temperature drops from 340K to below 320 K, I_G rapidly drops to below 5 nA.

We have also estimated the separate contributions of resistance variation due to doping and remnant ionic mobility with respect to temperature variation in the 340-320 K temperature range. Resistance variation due to eventual remnant ion mobility and doping has been measured to be well below 1 Ohm/minute at fixed temperature. In the same temperature range, for the different samples shown here resistance variation with temperature is in the order of 10-40 Ohm per minute for a temperature variation of 2K/min.

2. The presented phase diagram appears to point to the presence of a quantum critical point close to the optimal doping level. This is quite different from the bulk single crystal results that highlight a critical doping (if existing) close to 19 % and a pseudogap phase boundary of a possibly non-trivial form. Given the potential significance of this result and the uncertainty in the doping level estimation, related discussions are warranted.

Reply: In the following we strive to remain as factual as possible and avoid over-interpretation. Our data is different in another respect in that the phase boundary between the pseudogap part and the strange metal part is not linear. An extrapolation of the less than optimally doped part of the phase boundary would indicate a critical doping of closer to 19-20%. On the other hand the region of the phase boundary around the dome peak, despite the greater associated error, does indicate a critical doping close to the optimal value. We think that measurements on other systems and eventually at very high magnetic fields are needed before firm statements can be made in this regard. We have added the following text in the manuscript:

The extracted phase boundary, notably between pseudogap region and the strange metal region is not linear. Extrapolating this phase boundary to the doping axis would indicate a critical doping close to the optimal value, but higher values can be obtained by extrapolating segments of the phase boundary on the underdoped side.

3. In order to set up a benchmark for future studies using the same doping technique, it is necessary to provide more details and further elucidations about the technique. Whether or not some of them can be found in earlier papers by the same authors, it helps to make the paper self-contained.

3a. Is it purely electrostatic in nature? Any sign of irreversibility or hysteresis observed in experiment that might suggest a partial electrochemical character of the doping method?

Reply: The figure below on BSCCO, illustrates the reversibility of the method. The left panel shows the sequence of measurement from the initial doping (0) to the final doping (5). The right panels highlight the reversibility of the doping process by plotting T_c and the slope of the linear part of the resistivity curve as a function of doping which was changed in a non-monotonic manner. The coherence and monotonic nature of these quantities again validates the reversibility of the process. Similar measurements of reversibility have been obtained on graphene [Paradisi, A., et al. Applied Physics Letters **107**, 143103 (2015)] or MoS2 [Biscaras, J., et al. Nature Communications **6**, 1-8 (2015), *Supplementary information*] among others.

We have not observed any electrochemical character to date on any sample or measurement, but as we have mentioned earlier, going to the highest doping values in general degrades the sample near the contacts. We have added the following sentence in the manuscript and added the figure below and remarks above in Supplementary note 3.

We achieve a large, reversible variation in carrier density and hence in R_S and T_c by applying positive and negative V_g to our devices, accessing both underdoped and overdoped regions of the phase diagram (See Supplementary note 3).

3b. What is the thickness of the glass substrate? Why is the applied bias voltage so large compared to the liquid gating case? Is that because the glass substrate at high temperatures has a much worse metallicity than the ionic liquid?

Reply: The glass substrate is 0.5 mm thick essentially so as to provide mechanical stability. The applied bias voltage is high for both reasons: high 'viscosity' of ions in hot glass and the thickness of the glass substrate.

These details have been added to the supplementary note 1

3c. What is the effective thickness of the sample that has been charged for "thick" films (say >1 uc)? Since the screening length is not apriori known, this may bring in a big uncertainty in the data analysis. For example, for an overdoped cuprate, the whole film is conducting even without gating. Then after gating, its resistivity changes. But if we don't know the thickness of the film that has been charged, there is no way to separate the contributions of the charged and uncharged portions of the film and get a solid answer of how resistivity changes during gating.

Reply: Some literature does exist concerning screening length in cuprates for electrostatic doping. Smadici, S., et al., Phys. Rev. Lett. **102**, 107004 (2009) and Bollinger, A.T., et al. Nature 472, 458-460 (2011) indicate that in LSCO the screening length is sub nm, Mannhart, J., et al., Phys. Rev. Lett. **67**, 2099 (1991) and Frey, T. , et al., Phys. Rev. B **51**, 3257 (1995) indicate that in YBCO it is of the order of 1-1.5 nm, and that it could be twice as much in BSCCO. Our data on BSCCO is in good agreement with this estimation because sample A which is 2.5 nm thick shows a small discrepancy in the Hall coefficient estimated doping value. If a gradient in doping exists, it would show up in the superconducting transition width and it is indeed seen only in sample A. We remark in the manuscript: “device A (2.5 u.c.) can be expected to have a gradient in the doping over the total thickness”

We have added the above remarks to supplementary note 6 and referenced it in the article.

Reviewer #3 (Remarks to the Author):

In this manuscript, the authors used space charge doping (or others may call it solid *electrolyte gating*) to tune thin Bi2212 flakes with thickness from 1 to 2.5 unit cells. For samples with different doping level, they extracted characteristic temperatures from transport measurements and constructed a phase diagram. The research is highly interesting and valuable for people working on unconventional superconductivity and beyond. My comments are detailed below, and I would suggest a minor revision before the manuscript can be accepted for publication.

1. With sample B and C having the same thickness, the maximum attainable transition temperature varied by 20K, the authors suggested that it is due to effects of disorder, which is reasonable. Disorder seems also one of the reasons that they failed to observe superconductivity in samples with 0.5 unit cell thick. Another possible reason is the loss of carriers in such an ultra-thin sample after exfoliation, due to oxygen release or reconstruction in the Bi-O plane. It might be possible to apply different gate voltages and make measurements down to even lower temperature than 2.8K. Can the authors make such measurements or make comments on that?

Reply: We cannot but agree with the reviewer! At the moment we cannot make very low T measurements due to experimental constraints but we hope to do so in the medium term.

Both disorder and oxygen loss can result in T_c depression or suppression as we have stated in the manuscript. We have actually ‘tested’ both these possibilities, by either observing lower T_c’s on ‘rough’ substrates or by annealing 1 u.c. films at about 450K and rendering them gradually insulating. Bi-O plane reconstruction is however difficult to determine.

2. With electrolyte gating, it was already reported that small ions like Li⁺ or Na⁺ may get intercalated into gaps in the lattice (in this case it is the space between the Bi-O layers). Possible evidences include irreversible gating behaviour or sudden change of resistivity during gating. The authors shall give experimental evidences excluding this possibility, because otherwise, the interpretation of the data would become much more complicated.

Reply: Please refer to the reply of question 3a, Reviewer#2, which has been added to the supplementary material (Supplementary note 3) as proof of the reliability of the process.

3. As the authors have suggested, disorder plays an important role in such ultra-thin sample where

the exposed surface may become very reactive, ready to have physical or chemical absorption or subject to surface reconstruction. While 1 unit cell Bi2212 was reported to be insulating (ref. 18), and in this manuscript, it is superconducting. Is there any control of the sample storage or measurement environment? Did the authors observed any degradation in samples in less controlled environment?

Reply: The sample are exposed to moderately hot temperatures during the fabrication process and eventually during the contact deposition process. We strive to reduce this temperature as much as possible. Samples are stored in primary vacuum. Samples and devices A, B and D have been fabricated 2-4 days prior to insertion in high vacuum for measurement ($\sim 10^{-6}$ mbar). Sample C, which had to be stored for a month in primary vacuum, suffered an initial T_c degradation, probably due to oxygen loss. We have not made any systematic studies of survival or degradation in less controlled environments because in general a sample is measured till it 'dies'.

This information has been included in Supplementary note 4.

4. I would suggest that the authors add more data to table 1 to make it more complete. For example, initial T_c , maximum and minimum T_c and the gate voltages on which they were obtained. Although many of these information are available in figure 1, however, it would be good if the authors summarize these results quantitatively in the table.

Reply: This has been done and we thank the referee for suggesting this improvement.

Reviewers' comments:

Reviewer #1 (Remarks to the Author):

Review of Sterpetti et al.

I have gone through the detailed responses to all of the referee's comments, and find that the authors have satisfactorily addressed them, either by making additions to the manuscript, or presenting appropriate arguments deflecting the criticisms. After this reading of the manuscript, and the reviewer's comments an additional concern arose. This concern is not serious enough for me to recommend against publication, but it should be addressed briefly in the paper.

They indicate in the manuscript that there are two methods for determining the hole concentration, the use of the empirical formula, Equation 1, which relates the measured transition temperature to the number of holes per Cu, and the measured Hall coefficient $1/qRH$. The phase diagram in Fig. 3 is then determined using the empirical formula. The phase diagram in Fig. 4 is constructed using the Hall coefficient.

This concern has to do with the use of the Hall coefficient, $1/qRH$, to evaluate the carrier concentration, n_s . The authors remark on numbered page 4 that the Fermi surface and its evolution with doping are complex, and that the Hall coefficient is both temperature and doping dependent. They then assert that they will consider Hall coefficient values at the fixed temperature of 120 K.

They never justify the use of the Hall coefficient obtained at the fixed temperature of 120 K. To my knowledge the Hall coefficient accurately yields the number of holes per copper for some cuprates over particular ranges of hole concentrations and temperatures, and fails in this task outside of these ranges. They justify that two of their samples as prepared are at optimal doping, by their approximately equal Hall coefficients.

The authors should add some comment justifying the choice of 120 K as the temperature chosen for Hall measurements. They should also add some comment about the limitations of Fig.4, which is developed using the Hall coefficient. My suspicion is that a more elaborate analysis of the Hall effect and the transport resistance, as a function of temperature and doping than provided here, is needed to extract a truly meaningful phase diagram using the Hall coefficient.

Reviewer #2 (Remarks to the Author):

The authors have made adequate changes in response to the issues I previously brought up. The manuscript is now ready for publication.

Reviewer #3 (Remarks to the Author):

The authors have satisfactorily addressed all the points I raised, I have no more questions or comments. I suggest to accept it as it is.

Reply to reviewer #1

Our reply is in blue and in the revised manuscript the revisions have been signalled in red. We have also changed section titles and the order of end notes and the placement of the table as demanded in the editor checklist.

Reviewer #1 (Remarks to the Author):

I have gone through the detailed responses to all of the referee's comments, and find that the authors have satisfactorily addressed them, either by making additions to the manuscript, or presenting appropriate arguments deflecting the criticisms. After this reading of the manuscript, and the reviewer's comments an additional concern arose. This concern is not serious enough for me to recommend against publication, but it should be addressed briefly in the paper.

They indicate in the manuscript that there are two methods for determining the hole concentration, the use of the empirical formula, Equation 1, which relates the measured transition temperature to the number of holes per Cu, and the measured Hall coefficient $1/qRH$. The phase diagram in Fig. 3 is then determined using the empirical formula. The phase diagram in Fig. 4 is constructed using the Hall coefficient.

This concern has to do with the use of the Hall coefficient, $1/qRH$, to evaluate the carrier concentration, n_s . The authors remark on numbered page 4 that the Fermi surface and its evolution with doping are complex, and that the Hall coefficient is both temperature and doping dependent. They then assert that they will consider Hall coefficient values at the fixed temperature of 120 K.

They never justify the use of the Hall coefficient obtained at the fixed temperature of 120 K. To my knowledge the Hall coefficient accurately yields the number of holes per copper for some cuprates over particular ranges of hole concentrations and temperatures, and fails in this task outside of these ranges. They justify that two of their samples as prepared are at optimal doping, by their approximately equal Hall coefficients.

The authors should add some comment justifying the choice of 120 K as the temperature chosen for Hall measurements. They should also add some comment about the limitations of Fig.4, which is developed using the Hall coefficient. My suspicion is that a more elaborate analysis of the Hall effect and the transport resistance, as a function of temperature and doping than provided here, is needed to extract a truly meaningful phase diagram using the Hall coefficient.

We have now clarified both the choice of 120K temperature and the limitations of Fig4 in the appropriate places in the text and thank the reviewer for this suggestion.

120K: The Hall coefficient of cuprates indeed has a complex temperature dependent behaviour related to the complex electronic structure of cuprates. The Fermi surface evolves from hole like to electron-like and the contributions of these surfaces to the Hall coefficient are doping and temperature dependent. To minimise errors related to this dependence we consider R_H values at the fixed, low temperature of 120 K for all doping values. Several works (refs. 4,15,16,17,30) have elucidated this point. The choice of 120K is related to three criteria:

- 1) use a low temperature, above T_c , so that Hall coefficient changes due to a temperature induced change in the Fermi-Dirac distribution is minimised
- 2) use the same temperature for all samples and dopings
- 3) verify (figure S2a) that this choice is coherent.

We have modified the text as follows:

This value is indicative of the change introduced by doping and not an absolute measure of the carrier density since the electronic structure at the Fermi surface and its evolution with doping and temperature is complex and cannot be approximated by a single parabolic band. The Fermi surface evolves from hole-like to electron-like with doping and the contributions of these surfaces to the Hall coefficient are temperature dependent [15, 16, 30]. To minimise errors related to this dependence we consider R_H values at the fixed, low temperature of 120 K [4,17] for all doping values.

Limitations of Figure 4:

We have added the following text:

However we must keep in mind the limitations in the use of R_H for estimating doping discussed in section 1 above and note only the qualitative similarity between the phase diagrams of Figs. 3 and 4c.